# Calcium and Iron Content of Cereal-Based Gluten-Free Products

**DOI:** 10.3390/foods11142001

**Published:** 2022-07-06

**Authors:** Yvonne Jeanes, Ambra Spitale, Giorgia Nicolini, Voulla Bergmann, Lorretta Fagbemi, Rawan Rasheid, Camilla Hovland, Adele Costabile

**Affiliations:** School of Life and Health Sciences, University of Roehampton, London SW15 4JD, UK; ambra.spit@gmail.com (A.S.); giorgia.nicolini@student.univaq.it (G.N.); voulla.bergmann@roehampton.ac.uk (V.B.); fagbemil1@roehampton.ac.uk (L.F.); rawan_rashede@yahoo.com (R.R.); camillahov@gmail.com (C.H.); adele.costabile@roehampton.ac.uk (A.C.)

**Keywords:** calcium, iron, gluten-free, coeliac, bread, pasta

## Abstract

The impact of a gluten-free (GF) diet on the intake of calcium and iron is broadly unknown, as the micronutrient content of GF cereal-based products has scarcely been measured. The study aimed to measure the calcium and iron content of GF cereal-based products from the UK. Seventy-three GF products were analysed. A laboratory analysis of calcium and iron from GF food samples was performed by spectrophotometric and flame emission photometry, respectively. The values for wheat-based products were from a nutrient database. The calcium in GF white loaf samples varied greatly from 54 to 140 mg/100 g, with a lower average calcium content compared with wheat-based values (99 ± 29 mg/100 g *n* = 13 versus 177 mg/100 g; *p* < 0.01). Only 27% of the white loaves and rolls were fortified with calcium; this contrasts with 100% of white wheat-based loaves. The calcium in GF flour mixes ranged from 54 to 414 mg/100 g, with 66% fortified. GF white pasta had more calcium compared with wheat-based pasta (76 ± 27 mg/100 g *n* = 7 versus 24 mg/100 g; *p* = 0.002). The iron in GF bread loaves and pasta samples was similar to wheat-based comparators, whereas lower iron levels were observed in GF wraps (0.8 ± 0.2 *n* = 11 versus 1.6 mg/100 g). GF bread had a significantly higher fibre content, and the majority of GF bread had a lower protein content, compared with wheat-based bread products. These calcium and iron values provide a valuable addition towards enabling more accurate nutrient intake analysis for adults and children with coeliac disease.

## 1. Introduction

Globally, there has been a substantial rise in the number of people excluding gluten from their diet [1], with an associated growth in the manufactured gluten-free (GF) food market [2]. The growth of the GF food market is likely to be influenced by the current popularity of excluding gluten as a lifestyle choice for perceived health benefits, with endorsements by celebrities and sportspeople [3,4]. The strict exclusion of dietary gluten is required in the treatment of coeliac disease, dermatitis herpetiformis, gluten ataxia and gluten sensitivity. The prevalence of coeliac disease is approximately 1% worldwide [5].

Gluten is a storage protein found in wheat, barley and rye grains. Wheat flour is used ubiquitously in food staples such as breads and pasta. The GF food market includes alternatives to these wheat-based products, which are made with differing proportions of flours derived from maize, rice, potato, tapioca, pea and wheat-free starch [6,7]. Whilst the macronutrient compositions of GF and wheat-based foods have been reported, utilising mandatory data from manufacturer food labels [8,9], the micronutrient data are not mandatory and are predominantly absent from food labels. Likewise, the micronutrient data are predominantly missing from food and nutrient databases, as there has been very limited laboratory analysis of the micronutrient content of GF food products [10].

Iron and calcium are particularly relevant micronutrients for adults and children with coeliac disease. Increased bone fracture risk and osteoporosis are associated with coeliac disease [11]. There is increased prevalence of iron deficiency anaemia at the time of diagnosis with coeliac disease that can persist on the GF diet [12]. The measurement of the calcium and iron contents of GF cereal-based foods is of importance, as national dietary intake surveys from the United Kingdom, United States, Australia and Poland report that more than a quarter of ingested iron and calcium originate from cereals and cereal products [13,14,15,16]. It is the frequent consumption of cereal-based foods that results in their contribution to a relatively large proportion of calcium and iron intake. White wheat flour has been used as a vehicle for population-wide micronutrient fortification for decades, and is currently mandatorily fortified, with at least iron, in over 80 countries [17]. In the UK, but not the US, nor Canada, wheat flour is also mandatorily fortified with calcium [18]. Excluding wheat, as part of a GF diet, inadvertently results in adults and children with coeliac disease being excluded from the mandatory fortification programs.

Studies reporting the nutritional adequacy of diets consumed by adults and children with coeliac disease are inherently limited by the inadequate micronutrient data available for GF manufactured products [19,20]. In light of the paucity of data, the process used for calculating micronutrient data in these studies is not clear. Austrian and Italian researchers have each developed a theoretical nutrient database for GF foods based on calculations from the listed ingredients [21,22], with the need for direct measurements of micronutrients in GF products highlighted. A direct micronutrient analysis of a small selection of GF breads and pastas from Poland and Spain has been published [23,24]. As a consequence, determining the nutritional adequacy of the gluten-free diet of individuals and subpopulations remains elusive.

The calcium and iron contents of UK GF foods have not been directly measured. These values would be a significant data source when estimating the dietary intake of calcium and iron in people excluding gluten from their diet. The study aimed to determine the iron and calcium contents of a comprehensive sample of cereal-based GF products.

## 2. Materials and Methods

### 2.1. Gluten-Free (GF) and Wheat-Based Food Products

In 2019, 73 GF food products were sourced either directly from manufacturers or purchased from supermarkets within the UK. The following products were selected to provide a wide range of bread items and from discussions with a coeliac society regarding what would be useful for dietary analysis: 57 GF bread products, 7 GF pasta products and 6 multipurpose GF flour mixes (Appendix A). GF flour mixes include additional ingredients to improve the taste and texture of the baked food. Ingredients, energy (Kcal/100 g) and nutrient data were collated from food labels and manufacturer websites. Nutrient data per 100 g included: total fats (g); saturated fats (g); protein (g); total carbohydrates (g); sugars (g); total dietary fibre (g); and, where available, calcium (mg) and iron (mg). The energy and nutrient values of wheat-based foods were sourced from The Composition of UK Foods [25].

### 2.2. Quantitative Determination of Calcium by Jenway^TM^ PFP7 Flame Photometer

Calcium was extracted from GF samples with a lithium acetate solution and analysed directly by flame emission photometry (Jenway^TM^ PFP7 Flame Photometry, Cole-Parmer, Stone, UK). Food samples were homogenised using a food homogeniser and 1 g of each sample was transferred into 50 mL falcon tubes (Merck Life Science UK Limited, Gillingham, UK). Fifty millilitres of lithium acetate solution, prepared with 0.8 M/L lithium chloride (Merck Life Science UK Limited, Gillingham, UK) and 0.2 M/L lithium acetate (Merck Life Science UK Limited, Gillingham, UK) was added to the samples, and they were shaken vigorously overnight at room temperature, before filtering through a filter paper (Whatman No. 6, Merck Life Science UK Limited, Gillingham, UK) into a 500 mL volumetric flask. The filtrate was used to read with selection of the calcium filter position on flame emission photometry. The standard calibration method was used to prepare daily calibration curves (*n* ≥ 6) for each element. All calibration curves showed good linearity over the entire range of concentrations with acceptable quadratic correlation coefficients (R^2^ ≥ 0.999). All measurements were performed in triplicate. The final calcium concentration in all samples was calculated using the equation reported below:$$\mathrm{Ca}\;\left(\frac{\mathrm{mg}}{100\;g}\right)=\frac{\left(\mathrm{sample}\;\mathrm{conc}\;\left(\mathrm{ppm}\right)\times \;\mathrm{density}\;\mathrm{of}\;\mathrm{Ca}\right)}{20}\times 100$$

### 2.3. Spectrophotometric Determination of Iron

The spectrophotometric method [26] was performed with a Genesys 2 UV-VIS Spectrometer, Model TM2 (Mettler-Toledo Ltd., Leicester, UK). Aliquots of 2.5 g of each sample were placed in a crucible and converted to ashes in a hot air oven (100 °C) for approximately 7 h. After that, the samples were taken out from the container and subsequently transferred into a small glass beaker. For the determination of Fe(III) content, 10 mL of hydrochloric acid 2 mol/L solution and 2.5 mL of potassium thiocyanate solution (0.1 mol/L, Merck Life Science UK Limited, Gillingham, UK) were added to 10 mL of the sample. The sample was ready for spectrophotometric analysis after 10 min. The ionic iron was the sum of Fe(II) and Fe(III) contents. Before spectrophotometric analysis, intensity of colour was increased by addition of potassium thiocyanate (Merck Life Science UK Limited, Gillingham, UK) for complexation of iron ions and formation of red complex with different composition from Fe^3+^(aq) + SCN^−^(aq) → Fe(SCN)^2+^(aq). Standard stock of Fe(NO_3_)_3_ solutions was prepared by dissolving 0.004 of Fe(NO_3_)_3_ (Merck Life Science UK Limited, Gillingham, UK) in 9 mL of 0.1 mol/L in a volumetric flask (10 mL). The calibration solutions were prepared by pipetting volumes of 0, 0.50, 1.00, 1.50 and 2.00 mL, respectively, of the stock standard solution into volumetric flasks (10 mL). Next, 2.5 mL of potassium thiocyanate (0.1 mol/L) was added to each test tube to obtain a concentration range from 0 to 1 mmol/L. The absorbance of each solution (working and analysed solutions) was measured at a wavelength of 458 nm using a 10 mm quartz cuvette.

### 2.4. Statistical Analysis

All data were analysed using SPSS (version 26 IBM software, Armonk, NY, USA). A one-sample *t*-test was used to determine the significant differences between the average nutrient values of the GF products and the nutrient values for their wheat-based counterparts from The Composition of UK Foods [25]. The level of significance was set at *p* < 0.05.

## 3. Results

The energy, macronutrient, calcium and iron contents of 73 GF products are presented, inclusive of 57 GF bread products, 7 GF pasta products and 6 multipurpose GF flour mixes (Appendix A). All GF bread categories had significantly higher fibre contents compared with wheat-based bread products, and the majority of GF breads had lower protein contents (Table 1).

Substantial variation was observed within the nutrient composition of GF white bread loaves; the fibre content ranged from 1.7 g to 12.5 g/100 g and sugar from 0.1 g to 5.8 g/100 g. Maize starch and rice flour were the predominant ingredients for all GF breads, whilst some also included millet, buckwheat, tapioca, potato or quinoa flour (Appendix A). Only 27% of white loaves and rolls were fortified with calcium. This contrasts with the mandatory fortification of white wheat-based breads in the UK.

The average calcium content of GF white loaf samples was significantly lower than the wheat-based white loaf value (*n* = 12; *p* < 0.01), including those fortified with calcium (*n* = 5; *p* < 0.01) (Figure 1). The calcium content of GF white loaf samples varied greatly from 54 to 140 mg/100 g, with one outlier excluded from the analysis due to very high fortification (596 mg/100 g). The calcium content of GF white roll samples was significantly lower than wheat-based white rolls (*n* = 5; *p* < 0.01), and there were no calcium-fortified GF white rolls. The calcium contents of GF brown and seeded loaves were similar to the wheat-based version (Figure 1).

GF white pasta samples (*n* = 7) had significantly more calcium compared with wheat-based pasta (76 ± 27 mg/100 g versus 24 mg/100 g; *p* = 0.002, Table 1). The ingredients used included maize, rice, quinoa, sorghum, pea and lentil flours (Appendix A). The calcium values of the fortified GF flour mixes (*n* = 4) ranged from 119 to 414 mg/100 g (226 ± 165 mg/100 g). The calcium contents of the two GF flour mix products that were not fortified were 54 mg and 71 mg, respectively.

The iron content of the GF bread loaf samples was similar to the values reported for wheat-based products, whereas lower iron levels were measured in other GF bread product samples, for example, wraps (Table 1). The GF white pasta samples (*n* = 7) had similar amounts of iron (1.4 ± 1.3 mg/100 g and 1.6 mg/100 g, respectively; *p* = 0.75). The iron content of GF flour mixes (*n* = 6) was 1.6 ± 1.7 mg/100 g, and none were fortified with iron (Appendix A).

## 4. Discussion

This is the first study to present a direct laboratory analysis of calcium and iron from a comprehensive sample of GF products, enabling a greater understanding of the variability of calcium content in particular. Our data are unique in a large number of bread samples and subcategories, which is more aligned to nutrient databases.

A wide range of calcium and iron values was evident. This was likely due to inconsistent fortification of the nutrients and the variety of ingredients used within the GF products. The average calcium and iron contents of the UK GF breads we have reported were similar to the direct analysis of small samples of Spanish (*n* = 12) and Polish (*n* = 5) GF breads [23,24]. The average calcium value of GF pasta was higher than the Spanish (*n* = 12) and Polish (*n* = 4) GF pasta values; the large range of values was evident in all three studies. A limitation of our study is the use of published values for the wheat-based comparators, rather than conducting a laboratory analysis, and the small number of products in some subcategories. Our study reports a direct chemical analysis of calcium and iron, improving the very limited micronutrient data for GF products [27]. There is clear scope for a greater number of micronutrients to be analysed.

Maize and rice flours were the predominant ingredients within the GF bread products analysed (Appendix A). Similarly to a Spanish study [27], these are lower in calcium and iron than pseudocereals such as buckwheat and quinoa [28]. The inclusion of pseudocereals in GF breads is variable, which is of relevance as their inclusion can improve the micronutrient content [29]; for example, pea flour, as used in one of the pasta products, has a relatively high iron content [30].

Fortification of GF foods is voluntary, which has resulted in vastly different amounts; for example, with calcium, there was between 65 and 596 mg/100 g in the white GF bread loaf samples. This is relevant to clinical practice and assessing the nutritional adequacy of a patient’s GF diet. To illustrate this, if a person were to choose one rather than the other, two medium slices would provide 4% rather than 43% of their daily calcium requirement. Calcium fortification is only mandatory in white wheat flour in the UK, excluding all cereal-based GF products from mandatory fortification. This issue also covers the B vitamins and will include folic acid when this becomes mandatory in the UK. Of note, a higher proportion of prescribed GF white breads are fortified with calcium (53%), compared with those commercially available GF breads (16%), thus inequality in access to prescribed products could also impact calcium intake [31].

Until nutrient databases include the micronutrient content of GF products, it will remain challenging to establish if a person or subgroup of the population consuming a GF diet has low micronutrient intake. Further work is required to analyse a broader range of micronutrients in GF food products to provide the data required for such databases.

In agreement with previously published studies, the GF breads had predominantly lower protein and higher fibre contents compared with wheat-based breads [6,9,27]. This is due to the well-known lower protein content of many GF flours used and the addition of plant fibres [6].

## 5. Conclusions

These calcium and iron values provide a valuable addition towards enabling more accurate nutrient intake analysis for adults and children with coeliac disease. The biggest impact on calcium content was whether the product had been fortified. We recommend mandatory calcium fortification of all GF breads to promote adequate dietary calcium intake in a population with an increased risk of fractures. It is important that healthcare professionals are aware of the variability in calcium and iron and highlight this variability to patients.

## Figures and Tables

**Figure 1 foods-11-02001-f001:**
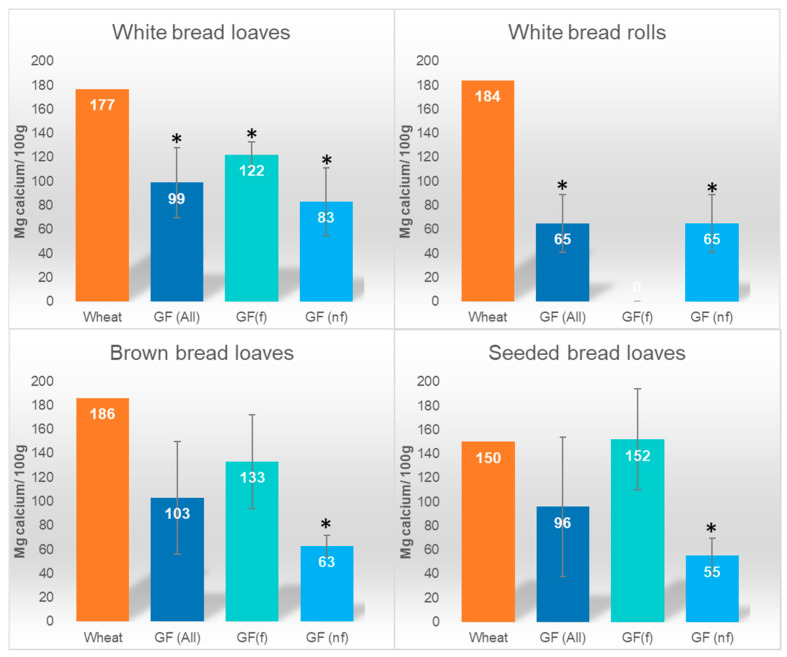
Calcium content of wheat-based and gluten-free (GF) bread products. GF(f): calcium-fortified products, GF(nf): no calcium fortification; calcium fortification category based on information from food label and ingredient list. * *p* < 0.05 compared with wheat-based bread value.

**Table 1 foods-11-02001-t001:** Mean nutrient composition of gluten-free and wheat-based breads and pastas per 100 g.

	Energy (kcal)	Total Fat (g)	SFA (g)	CHO (g)	Sugars (g)	Fibre (g)	Protein (g)	Iron (mg)	Calcium (mg)
White GF loaves (*n* = 13)	229 ± 18 *	3.6 ± 1.6 *	0.4 ± 0.2 *	41.2 ± 5.3 *	1.9 ± 1.7 *	5.9 ± 3.1 *	4.5 ± 1.3 *	1.5 ± 0.5	^a^ 99.4 ± 29.2 *
White wheat-based loaves	219	1.6	0.29	46.1	3.4	2.5	7.9	1.6	177
Brown GF loaves (*n* = 7)	235 ± 16 *	4.4 ± 1.5 *	0.6 ± 0.2	41.0 ± 1.8	3.0 ± 1.9	7.3 ± 1.6 *	4.1 ± 1.1 *	2.2 ± 0.5	103 ± 46.7 *
Brown wheat-based loaves	207	2.0	0.43	42.1	3.4	5.0	7.9	2.2	186
Seeded GF loaves (*n* = 7)	257 ± 15	7.6 ± 1.1	1.0 ± 0.2	36.4 ± 5.0 *	2.7 ± 1.7	9.6 ± 2.3 *	5.1 ± 1.7 *	1.7 ± 0.7	96.4 ± 58.3
Seeded wheat-based loaves	270	7.4	1.0	43.8	3.8	6.2	9.9	2.3	150
White GF rolls (*n* = 5)	245 ± 20	3.0 ± 0.3 *	0.5 ± 0.2	47.5 ± 5.7	4.4 ± 1.3 *	6.7 ± 1.2 *	3.5 ± 0.8 *	1.4 ± 0.5	65 ± 24 *
White wheat-based rolls	254	2.6	0.62	51.5	2.6	2.6	9.3	1.5	184
Brown GF rolls (*n* = 3)	263 ± 18	7.2 ± 1.2 *	0.9 ± 0.3	38.7 ± 2.8	3.9 ± 1.4	8.1 ± 0.7 *	6.8 ± 1.1	1.1 ± 0.1 *	53.6 ± 26.9 *
Brown wheat-based rolls	236	3.2	1.1	44.8	2.8	4.3	9.9	2.4	201
White GF baguette (*n* = 4)	261 ± 10	3.5 ± 1.5	0.5 ± 0.2	51.0 ± 2.3 *	3.9 ± 1.6	5.9 ± 0.9 *	3.4 ± 0.7 *	0.9 ± 0.3 *	73.7 ± 14.6 *
White wheat-based baguette	263	1.9	0.34	56.1	2.8	3.3	9.0	1.4	121
GF wraps (*n* = 11)	234 ± 42 *	4.3 ± 1.7 *	0.5 ± 0.2 *	37.8 ± 10.9 *	3.0 ± 2.2	12.3 ± 3.8 *	5.9 ± 4.0	0.8 ± 0.2 *	108.9 ± 16.9 *
White wheat-based wraps	285	5.7	2.5	53.9	2.0	3.6	7.8	1.6	148
GF naan bread (*n* = 3)	288 ± 22	8.0 ± 1.3	1.0 ± 0.1	47.7 ± 4.0	1.8 ± 1.2	6.9 ± 1.2 *	2.8 ± 0.4 *	0.5 ± 0.2 *	113.0 ± 14.8 *
Wheat based naan bread	285	7.3	0.97	50.2	3.1	2.9	7.8	1.6	187
GF Pitta bread (*n* = 2)	222	3.5	0.4	38.3	4.6	10.9	3.8	0.8 (*n* = 1)	106.0
Wheat based pitta bread	255	1.3	0.2	55.1	3.0	2.3	9.1	1.9	138
GF Pizza base (*n* = 2)	276	3.9	0.6	53.6	4.5	7.3	2.	0.7	98.5
Wheat based pizza base	290	4.8	N	57.5	3.4	2.5	7.8	1.6	86
GF dry white pasta (*n* = 7)	244 ± 103 *	1.1 ± 05 *	0.3 ± 0.2	49.4 ± 24.8 *	1.0 ± 1.5	3.1 ± 2.4	7.6 ± 3.5 *	1.4 ± 1.3	76 ± 27 *
Wheat-based dry white pasta	343	1.6	0.23	75.6	2.1	N	11.3	1.59	24

Mean ± Standard Deviation for Energy and macronutrient data for GF (gluten-free) products from their labels. Calcium and iron measured in the laboratory. ^a^ *n* = 12 (excluded an outlier due to substantially higher calcium fortification, value was 596 mg/100 g). Wheat-based breads and pasta data [25], N: no value available for dry pasta. * Statistical difference between nutrient values of GF and wheat-based products determined by one-sample *t*-test, when 3 or more values available for GF product. Gluten-free products are marked in grey.

## Data Availability

Data is contained within the article or Appendix A.

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
