# Peer review of "Calcium and Iron Content of Cereal-Based Gluten-Free Products"

_foods, 2022, doi:10.3390/foods11142001_

Round 1
Reviewer 1 Report
Comment to authors:
Jeanes et al. Investigated the calcium and iron content of 73 gluten free cereal product from UK market collected in 2019. The manuscript brings many useful information for scientists and producers. Yet, the data are poorly presented, and the manuscript contains only one figure with the results. Thus, the gathered data should be better presented. The references style should be adjusted to the recommended journal style. More specific comments are given bellow.
I suggest modifying the title e.g. Calcium and iron content of cereal-based gluten-free products from UK market
Reference 26 is Statista not Statistica
Lines 98-99: Please check the temperature of ashing; 100°C seems rather low temperature for ashing
Line 119: please place the significance level within the previous sentence where appropriate
Line 127: I do not see Table 1, and I suggest preparing the plot of Fe content depending on the analysed product type (e.g. bread, pasta, breakfast cereal etc.) and broaden the discussion on iron
Line 138: Compared to what?
Line 139-140: It is not clear, do you mean also brown products?
Lines 141-142 incomplete sentence, I could not understand the meaning
Fig 1: I suggest adding the data also for flour mixes and pasta; please assure the higher resolution for print
Line 153-154: Please describe the difference between flour mixes and flour mix products for broader audience
Supplementary table: Please try to improve consistency in reporting the flour type: e.g. somewhere is written just rice, and for other products refined or wholegrain rice. The ingredients listed are only flours; please list also additional ingredients (such as bakery fats, hydrocolloids, protein concentrates, etc.) which also affect the nutritive value of the product. Moreover, I suggest to include the frequency plot of ingredients in the manuscript (at least flours or structuring ingredients such as hydrocolloids) in different GF product: bread, rolls, pasta, etc. It is not clear if iron content was measured or taken from the label, if measured please provide also standard deviation along with mean values (also for calcium). Please add the information of sodium or salt in products if possible (from the label). Please describe the abbreviations and be aware of the typing error for potato (20).
Author Response
Response to reviewer 1
Thank you for your time and constructive comments, please see our responses below:
Jeanes et al. Investigated the calcium and iron content of 73 gluten free cereal product from UK market collected in 2019. The manuscript brings many useful information for scientists and producers. Yet, the data are poorly presented, and the manuscript contains only one figure with the results. Thus, the gathered data should be better presented. The references style should be adjusted to the recommended journal style. More specific comments are given bellow.
I suggest modifying the title e.g. Calcium and iron content of cereal-based gluten-free products from UK market
Thank you for your suggestion, we have added this information to the abstract :The study aimed to measure the calcium and iron content of GF cereal-based products from the UK
Reference 26 is Statista not Statistica corrected.
Lines 98-99: Please check the temperature of ashing; 100°C seems rather low temperature for ashing We can confirm this is the correct temperature we undertook for ashing.
Line 119: please place the significance level within the previous sentence where appropriate
Modified to: The level of significance was set at P < 0.05.
Line 127: I do not see Table 1, and I suggest preparing the plot of Fe content depending on the analysed product type (e.g. bread, pasta, breakfast cereal etc.) and broaden the discussion on iron
In error table 1 was excluded from the original submission, table 1 now included within the manuscript
Line 138: Compared to what?
Modified to “The calcium content of GF white roll samples were significantly lower than wheat-based white rolls (n=5; p<0.01),”
Line 139-140: It is not clear, do you mean also brown products? & Lines 141-142 incomplete sentence, I could not understand the meaning
Modified to “ The calcium content of GF brown and seeded loaves were similar to the wheat-based versions (Figure 1).”
Fig 1: I suggest adding the data also for flour mixes and pasta; please assure the higher resolution for print
Table 1, now included within the manuscript, includes the data for pasta. Data for flour mixes is within the text lines…
Line 153-154: Please describe the difference between flour mixes and flour mix products for broader audience Included “GF flour mixes include additional ingredients to improve taste and texture of the baked food.”
Supplementary table: Please try to improve consistency in reporting the flour type: e.g. somewhere is written just rice, and for other products refined or wholegrain rice. The ingredients listed are only flours; please list also additional ingredients (such as bakery fats, hydrocolloids, protein concentrates, etc.) which also affect the nutritive value of the product.
Thank you, we recognise the inconsistency in data presentation, the data is taken from the food labels which varied in the detail provided. We have added the source of information within the supplementary table legend. The focus is on the measured Ca and Fe, we have included whether the foods were fortified with these micronutrients.
Moreover, I suggest to include the frequency plot of ingredients in the manuscript (at least flours or structuring ingredients such as hydrocolloids) in different GF product: bread, rolls, pasta, etc. Thank you for this suggestion, two recent reports have included this analysis (Allen and Orfila (2018) and Calvo-Lerma et al (2019)) so we suggest it would add little new information for the readers.
It is not clear if iron content was measured or taken from the label, if measured please provide also standard deviation along with mean values (also for calcium). Ca and Fe were measured in the laboratory for each product individually, this fact is now included within the table legend.
Please add the information of sodium or salt in products if possible (from the label). This is a good point, however, the focus of the manuscript is the novel aspect of laboratory measured amounts of calcium and iron and the sodium content from food labels has been published recently (Fry et al 2017).
Please describe the abbreviations Added to the supplementary table legend
typing error for potato (20). Thank you, corrected
Reviewer 2 Report
The manuscript “Calcium and iron content of cereal based gluten-free products” claims to report the main findings of a study investigating calcium and iron content in a selection of gluten-free (GF) food products. The aim of this study is of paramount importance in the field of GF products characterization.
Apart from inaccuracy related to the Reference style, the paper lacks a sound organization in the presentation of results. The Authors should revise the database and present results in a more harmonized and consistent way. Please, find below some inputs for revision.
General
Line 2: Title: please, delete the final full-stop
Lines 3-4: Author affiliations are missing. Please, check the Journal guidelines.
References within the text: please check the Journal guidelines and have the style of references within the text matching the Journal style, that is numbers within squared brackets [1], [2], etc.
Related to this issue, please, revise extensively the Reference list. It was not inserted according to the Journal style. Check Journal guidelines and amend the list accordingly.
Introduction
Lines 22-24: please, add a reference for the first statement to provide evidence of the increase in the manufacturing of gluten-free food products.
I suggest revising the Introduction, as at the present state it sounds a bit disjointed. Sentences should be organized in a more harmonized way, with a more consistent logical flow.
Materials and Methods
Line 72: I suggest adding a reference to the list of 73 food products available in the Supplementary file.
For the sake of clarity, I also suggest providing some details about the different groups of GF food products (i.e. a total of XX bread, a total amount of XX pasta products, etc.) here and not in the results section.
Results
This section should be organized in a more sound way. When you provide data for DF and protein, for instance, do not limit to bread. As you have at least three categories of food products (bread products, pasta products and flour mixes), DF and protein content should be presented and commented for all categories. Please, mind that you also have 1 biscuit and flaked cereals (for which I would recommend collecting more data). In addition, why did you limit the presentation of results for DF, sugars and protein? Fats are another big issue in GF bread, in particular. So they worth being discussed.
The same lack of soundness in data presentation exists for calcium and iron. Be more precise. Do not jump from data about iron to data about calcium.
Figures: why are you plotting only data about bread? What about pasta? Flour mixes? In addition, if you provide a figure with data splitted for each type of bread (which is good), you should also provide all data for these categories
Lines 145-147: please check the Journal style for Figures footnotes.
Please, mind about the limitations of the study the very low number of food items.
Discussion
Line 173: please, provide a reference.
Discussion should be more comprehensive and should include comparisons with more data available in literature.
Author Response
Response to reviewer 2
Thank you for your time and detailed constructive comments, please see our responses below:
The manuscript “Calcium and iron content of cereal based gluten-free products” claims to report the main findings of a study investigating calcium and iron content in a selection of gluten-free (GF) food products. The aim of this study is of paramount importance in the field of GF products characterization.
Apart from inaccuracy related to the Reference style, the paper lacks a sound organization in the presentation of results. The Authors should revise the database and present results in a more harmonized and consistent way. Please, find below some inputs for revision.
General
Line 2: Title: please, delete the final full-stop corrected
Lines 3-4: Author affiliations are missing. Please, check the Journal guidelines. This information has been provided and I understand the journal will include it into the manuscript
References within the text: please check the Journal guidelines and have the style of references within the text matching the Journal style, that is numbers within squared brackets [1], [2], etc. Related to this issue, please, revise extensively the Reference list. It was not inserted according to the Journal style. Check Journal guidelines and amend the list accordingly.
The manuscript was submitted as per Free Format Submission, the referencing has now been formatted to the ACS MDPI format and author affiliations are included in the word document
Introduction
Lines 22-24: please, add a reference for the first statement to provide evidence of the increase in the manufacturing of gluten-free food products.
Added: Lerner, B.A., Green, P.H.R. & Lebwohl, B. Going Against the Grains: Gluten-Free Diets in Patients Without Celiac Disease—Worthwhile or Not?. Dig Dis Sci 64, 1740–1747 (2019). https://doi.org/10.1007/s10620-019-05663-x
I suggest revising the Introduction, as at the present state it sounds a bit disjointed. Sentences should be organized in a more harmonized way, with a more consistent logical flow. The introduction has been reviewed with an improved logical flow of information.
Materials and Methods
Line 72: I suggest adding a reference to the list of 73 food products available in the Supplementary file. The following has been added to the methods: “57 GF bread products, 7 GF pasta products and 6 multipurpose GF flour mixes (supplementary data table).”
For the sake of clarity, I also suggest providing some details about the different groups of GF food products (i.e. a total of XX bread, a total amount of XX pasta products, etc.) here and not in the results section. Included “GF flour mixes include additional ingredients to improve taste and texture of the baked food.”
Results
This section should be organized in a more sound way. When you provide data for DF and protein, for instance, do not limit to bread. As you have at least three categories of food products (bread products, pasta products and flour mixes), DF and protein content should be presented and commented for all categories. Please, mind that you also have 1 biscuit and flaked cereals (for which I would recommend collecting more data). In addition, why did you limit the presentation of results for DF, sugars and protein? Fats are another big issue in GF bread, in particular. So they worth being discussed. These are good points, in error the ‘table 1’ was omitted from the word documents of the original submission. Table 1 includes data on Fats, DF and protein in more details. We note the importance of macronutrient differences between products, the novel focus of the paper is on laboratory measured calcium and iron. We recognise the limited data on flaked cereal and breakfast biscuit, thus this is in the supplementary data and not a focus of the manuscript.
The same lack of soundness in data presentation exists for calcium and iron. Be more precise. Do not jump from data about iron to data about calcium. We have reviewed the flow of information within the results, taking into consideration of inserting table 1 and have improved the flow of information presented.
Figures: why are you plotting only data about bread? What about pasta? Flour mixes? In addition, if you provide a figure with data splitted for each type of bread (which is good), you should also provide all data for these categories.
Thank you for this observation, we agree and have this data presented within table 1.
Lines 145-147: please check the Journal style for Figures footnotes.
The guidance has been reviewed an meets that of “All Figures, Schemes and Tables should have a short explanatory title and caption”
Please, mind about the limitations of the study the very low number of food items.
This is addressed within the discussion, “…and the small number of products in some sub categories”
Discussion
Line 173: please, provide a reference. Added the following reference which addresses this point: Calvo-Lerma, et al. Differences in the macronutrient and dietary fibre profile of gluten-free products as compared to their gluten-containing counterparts. Eur. J. Clin. Nutr. 2019, 73, 930-936
Discussion should be more comprehensive and should include comparisons with more data available in literature. This is a good point and highlights precisely why there is a need for the current study as there is extremely limited data available on the micronutrient composition of GF foods.
Reviewer 3 Report
This is a well-written paper with relevant information.
Major improvements are required in the presentation of all the data now shown in one table, the Supplementary data table (SDT) .
The text in the Results section refers to this table and to Table 1. Table 1 is not present.
Results in the SDT are presented per xxx gram (should be mentioned.
Explanation of abbreviations is missing.
I recommend that this additional information should be mentioned in a text below the SDT. Also other information should be mentioned in such a text: for instance that data for Ca and Fe are measured data, the others are found in the database mentioned also in the text of the paper.
I also recommend to include this table in the text of the paper itself, since it includes essential information.
In addition to data of Ca and Fe, the shown data of macronutrients (e.g. protein, fibre). is valuable information. The presence of these data should be mentioned in the abstract. Otherwise they remain too invisible. No propblem for me to extend the abstract to220-230 words.
Author Response
Response to reviewer 3
Thank you for your time and constructive comments, please see our responses below:
This is a well-written paper with relevant information.
Major improvements are required in the presentation of all the data now shown in one table, the Supplementary data table (SDT) . There was an error in the first submission in that the submitted table 1 did not appear within the manuscript
The text in the Results section refers to this table and to Table 1. Table 1 is not present.
Table 1 now included within the manuscript
Results in the SDT are presented per xxx gram (should be mentioned.
Thank you, Table title modified to: Energy and nutrient data for individual gluten-free products per 100g, inclusive of predominant flour types within the product.
Explanation of abbreviations is missing.
Than you, included for table legend: Predominate flour type, energy and macronutrient data for GF products from their labels. Calcium and Iron measured in the laboratory. SFA; saturated fatty acids, CHO; carbohydrate, F; fortified,
I recommend that this additional information should be mentioned in a text below the SDT. Also other information should be mentioned in such a text: for instance that data for Ca and Fe are measured data, the others are found in the database mentioned also in the text of the paper.
Thank you, included for table legend: Predominate flour type, energy and macronutrient data for GF products from their labels. Calcium and Iron measured in the laboratory. SFA; saturated fatty acids, CHO; carbohydrate, F; fortified,
I also recommend to include this table in the text of the paper itself, since it includes essential information. This is an interesting point, we understand the journal style is for the data to be presented as supplementary information.
In addition to data of Ca and Fe, the shown data of macronutrients (e.g. protein, fibre). is valuable information. The presence of these data should be mentioned in the abstract. Otherwise they remain too invisible. No problem for me to extend the abstract to220-230 words. The following has been added to the abstract: “GF bread had significantly higher fibre content, and the majority of GF breads had lower protein content, compared with wheat-based bread products.”
Round 2
Reviewer 1 Report
Authors have improved their manuscript to satifactory level, I do not have more comments.
Author Response
Thank you for your time and comments. Most appreciated.
Reviewer 2 Report
Authors amended the manuscript according to previous suggestions/comments only in a few points. The scientific flaws identified in the first round review thus remain.
The paper mainly lacks a good presentation and discussion of results, which hampers the paper to be considered suitable for publication.
Author Response
We thank you for taking the time to read our detailed responses to all of the points you raised in your detailed first review.
Our point by point response to your briefer second review:
Authors amended the manuscript according to previous suggestions/comments only in a few points. The scientific flaws identified in the first round review thus remain.
We addressed every point raised in the first review. The manuscript focuses on addressing the study aim of ‘determine the iron and calcium content of a comprehensive sample of GF products.’ This has been comprehensively addressed, with data relevant to the study aim presented to a good standard and discussed in context of the very limited published studies in this area.
We are concerned about the comment of ‘scientific flaws’ which is not mentioned in your first comprehensive review.
The paper mainly lacks a good presentation and discussion of results, which hampers the paper to be considered suitable for publication.
Our manuscript focus and novelty is the laboratory measured calcium and iron analysis which are clearly presented within the text, table one, figures and within the supplementary data table.
Greater detail of macronutrient content from the food labels which would detract away from the main focus of the manuscript, the macronutrient content of GF foods from food labels has been reported fairly recently and would add little new information to the manuscript (Fry et al 2017).
We addressed the comment about discussion by “Discussion should be more comprehensive and should include comparisons with more data available in literature.” However, as it was raised again we have added some more to the discussion, but see little benefit in discussing the macronutrient content in greater detail as this is not the focus of the paper – we wished to highlight that the macronutrient profiles were indeed similar to published studies.
Reviewer 3 Report
I agree with the adaptations in the manuscript.
One item in the now added Table 1 should be clarified - "fibre by AOAC method" not clear, since there are a range of AOAC fibre methods: DOI: 10.1016/j.foodchem.2012.09.029
The most important are the two widely used 'classical' methods AOAC 985.29 and AOAC 991.43, .AOAC 2011.25 (and 2009.01) are more recent - they measure also low MW fibre - so (mostly somewhat) higher fibre levels are found. See e.g. Dietary fibre: challenges in production and use of food composition data: DOI 10.1016/j.foodchem.2012.09.029. I presume that all or most fibre measurements were done with 985.29 / 991.43
Please specify the AOAC method(s) used
Author Response
Thank you for this observation, I see this is of potential confusion for the readers and thus removed reference to AOCA method as we did not measure the fibre content, this data was taken from The Composition of UK Foods [25] for wheat based foods and food labels with unknown method for GF foods.